# ADST: Forecasting Metro Flow Using Attention-Based Deep Spatial-Temporal Networks with Multi-Task Learning

**DOI:** 10.3390/s20164574

**Published:** 2020-08-14

**Authors:** Hongwei Jia, Haiyong Luo, Hao Wang, Fang Zhao, Qixue Ke, Mingyao Wu, Yunyun Zhao

**Affiliations:** 1School of Software Engineering, Beijing University of Posts and Telecommunications, Beijing 100876, China; jiahongwei@bupt.edu.cn (H.J.); wh1995@bupt.edu.cn (H.W.); zfsse@bupt.edu.cn (F.Z.); kqx@bupt.edu.cn (Q.K.); 2019140679@bupt.edu.cn (M.W.); knot_zyy@bupt.edu.cn (Y.Z.); 2Beijing Key Laboratory of Mobile Computing and Pervasive Device, Institute of Computing Technology Chinese Academy of Sciences, Beijing 200190, China

**Keywords:** forecasting passenger flow, spatiotemporal networks, multi-task learning, attention mechanism, spatiotemporal dependency

## Abstract

Passenger flow prediction has drawn increasing attention in the deep learning research field due to its great importance in traffic management and public safety. The major challenge of this essential task lies in multiple spatiotemporal correlations that exhibit complex non-linear correlations. Although both the spatial and temporal perspectives have been considered in modeling, most existing works have ignored complex temporal correlations or underlying spatial similarity. In this paper, we identify the unique spatiotemporal correlation of urban metro flow, and propose an attention-based deep spatiotemporal network with multi-task learning (ADST-Net) at a citywide level to predict the future flow from historical observations. ADST-Net uses three independent channels with the same structure to model the recent, daily-periodic and weekly-periodic complicated spatiotemporal correlations, respectively. Specifically, each channel uses the framework of residual networks, the rectified block and the multi-scale convolutions to mine spatiotemporal correlations. The residual networks can effectively overcome the gradient vanishing problem. The rectified block adopts an attentional mechanism to automatically reweigh measurements at different time intervals, and the multi-scale convolutions are used to extract explicit spatial relationships. ADST-Net also introduces an external embedding mechanism to extract the influence of external factors on flow prediction, such as weather conditions. Furthermore, we enforce multi-task learning to utilize transition passenger flow volume prediction as an auxiliary task during the training process for generalization. Through this model, we can not only capture the steady trend, but also the sudden changes of passenger flow. Extensive experimental results on two real-world traffic flow datasets demonstrate the obvious improvement and superior performance of our proposed algorithm compared with state-of-the-art baselines.

## 1. Introduction

With the rapid development of modern transportation, urban traffic congestion has become a commonplace phenomenon, which has brought a series of travel safety problems. Nowadays, more and more travelers are choosing underground transportation so as to avoid congestion on the ground. In this case, metro flow prediction becomes an indispensable building component in data-driven urban computing to provide future reliable traffic guidance information, which will facilitate urban safety and running efficiency of its transportation systems (ITS) [1]. If metro conditions, especially the coming of burst flows, could be forecasted accurately, we would be able to respond ahead of time to prevent stations from experiencing congestion and acquire the crowdedness-aware route recommendation for the travel time reduction [2]. Therefore, a robust and accurate model is advised to be the basis of predicting urban metro flow to provide valuable warnings and deploy site security measures as early as possible. In this way, the city could mitigate congestion occurrence and improve metro utilization by means of big data and artificial intelligence technology [3].

The relevant literature denotes a long line of studies for researchers and practitioners in the field of traffic prediction including crowd flows [4], traffic volume [5], vehicle speed [6,7], taxi pick-ups [8], and travel time estimation [9], etc. Many attempts have been made on the traditional time series prediction algorithm for people’s individual trajectory data. AutoRegressive Integrated Moving Average (ARIMA) [10] and its variants [11,12] are used in early traffic forecasting works. Recent studies attempt to use spatial relationships [12,13,14] and external context data (e.g., weather, events) [15,16] for flow prediction. Duan et al. [12] proposed a spatial-temporal ARIMA (STARIMA) model to integrate spatial and temporal information for traffic flow prediction. The model can obtain better prediction accuracy than the traditional ARIMA model by using the spatial information of neighboring links. Although these studies confirm that prediction can be improved by considering various external factors (e.g., weather, holiday), they are still subject to complex nonlinear spatial-temporal correlations and show less practicality for fast online services (e.g., Amap, Baidu Map).

Deep learning has made a huge progress on many prevalent learning tasks [17]. Amazing results have inspired the field of traffic flow prediction, which represents a typical spatiotemporal prediction problem. For example, quite a few studies [18,19] modelled traffic grid data as a heat map image and used the Convolutional Neural Network (CNN) to extract the adjacent spatial dependency. To model complex temporal correlations, researchers utilize the Recurrent Neural Network (RNN) [20,21] to improve flow prediction accuracy. Some works [22,23] combined the CNN and Long Short-Term Memory (LSTM) to capture both the spatial and temporal correlations. Yao et al. also propose a method to jointly train the model by integrating local CNN and LSTM [8]. For graph-structured data, Guo et al. combine a traditional CNN with graph convolution to address this issue [5].

These pioneering attempts illuminate their superior performance compared with previous methods. However, the existing methods have some limitations. On one hand, the spatial properties between locations only rely on the neighbors from historical traffic [4,8] and the model could only learn an adjacent spatial dependency, but the correlations between remote locations also affect the traffic flow over time. On the other hand, different periodical time windows along historical time series, and even various time slices inside the window, would pose different impacts on the current time frame. Although previous studies have considered periodicity, sometimes they expose inadequate modeling or unsatisfactory prediction performance due to their hypotheses regarding strictly cyclical patterns in multiple scenarios.

To address the challenges described above, we first justify the problem with the real-world metro card swiping record in Figure 1 and find that the amount of check-in (or check-out) swiping of a specific station differs significantly as time goes by. The trend among weekdays with two rush hours is much more similar to each other, which is clearly separated from the weekends. Flows of stations located in different functional areas are also different. The flows in the business region are relatively higher than those in the attractions on weekdays, while the flows present the opposite results on weekends; the flows in the railway station are the largest on all of these days, especially the first day due to the New Year. Moreover, the rush hours on weekdays usually occur in the late afternoon, but would fluctuate from 17:30 p.m. to 18:00 p.m. on different days as the blue markers show in Figure 1.

Therefore, we propose a novel neural network named the attention-based deep spatial-temporal network with multi-task learning architecture (ADST-Net) for traffic flow prediction. ADST uses a residual neural network to learn a representation that effectively obtains spatiotemporal flow knowledge as well as transfer volume across stations. An attention mechanism is introduced to learn the temporal dependency, and a multi-scale CNN is leveraged to handle the latent spatial similarity. The global features (e.g., weather) and event features (e.g., holiday) are encoded as context information to improve the model. Meanwhile, a multi-task learning strategy regarded another flow property, e.g., the transition flow between two stations, as the additional task to produce more meaningful factors.

We conduct extensive experiments on two large-scale public datasets including MetroHZ and TaxiBJ. Both datasets are the discrete measurements recorded at fixed space locations at fixed time intervals. MetroHZ contains 70 million records of metro flow, ranging from 1 January 2019 to 25 January 2019, including 81 stations in total within three lines of Hangzhou. TaxiBJ consists of tremendous amounts of trajectories, which represent the taxicab GPS data and meteorology data in Beijing for four time periods: 1 July 2013–30 October 2013, 1 March 2014–30 June 2014, 1 March 2015–30 June 2015, 1 November 2015–10 April 2016. These trajectories can be used to calculate the traffic flow in each region. All erroneous and missing data are properly remedied. To fully evaluate the general ability of our proposed algorithm on different transportation patterns, we use the metro dataset of Hangzhou City and the popular public TaxiBJ dataset of Beijing City. As comparison, we also implement several state-of-the-art approaches.

The main contributions in this paper are summarized as follows:We propose a novel algorithm frame that jointly considers the spatial, temporal, and several aspects of contextual information (e.g., holidays, weekdays, weather, sluice machine account, functional region, transfer identifier) for the traffic passenger flow prediction. All these measurements are able to reflect the metro conditions together and accurately capture the passenger flow changes. The prediction benefits from our extracted factors and has an average of 38.29 mean absolute error (MAE) points lower than the baselines.By introducing a gated attention mechanism for fine-grained temporal modeling and a multi-scale strategy for latent spatial similarity into the residual network, our proposed model can capture not only the daily flow trends, but also sudden flow changes (e.g., Spring Festival) of all the stations. The attention mechanism is used to emphasize the effect of critical temporal periods with higher temporal dependencies for the flow prediction.A multi-task learning is introduced to improve flow prediction accuracy. The station flow (e.g., incoming and outgoing) prediction is modelled as the main task, and the transition flow forecasting is utilized as an auxiliary task to conveniently express more meaningful and general representation to boost the prediction performance.

## 2. Related Works

The topic of how to utilize big data to better predict traffic flow has become a fundamental problem in urban computing in recent years, and there exist plenty of works in this field. In the following, we will review the related literature on traffic passenger flow forecasting.

Traditional time series processing approaches, such as ARIMA, Kalman filtering, are widely applied in the traffic predicting system [10,12,24] under the consideration that the formation and dispersion of traffic flows are gradual. Recent studies explore the external context data such as area types, weather conditions, and event information to improve prediction performance [25,26]. Spatial measurements are modeled in several studies [27,28]. In general, these straightforward approaches use the raw data or handcrafted features for flow prediction, which lacks the ability to model complex latent spatial, temporal, and external properties simultaneously.

By extracting the nonlinear spatiotemporal dependencies, deep learning techniques have achieved great successes in computer vision and natural language processing [17]. Inspired by these progresses, a series of studies are proposed for traffic flow prediction based on deep learning techniques. Early studies stack several fully connected layers to fuse context data from multiple sources to predict traffic demand [29] and the taxi supply-demand gap [30]. These methods use intricate features through hierarchical representations, but only capture partial properties and neglect the complex spatial correlations and temporal interactions explicitly.

One line of classical studies employs CNN to encode complex spatial correlations. These methods treat the entire city’s traffic flows as an image or a matrix whose entries correspond to the flows of individual rectangular grid regions and the spatial structure is Euclidean. For instance, Zhang et al. (2017) [4] and Ma et al. (2017) [19] utilize CNN for flow forecasting. Considering that the traffic conditions are limited on road network graphs, some latest studies formulate the traffic flow prediction as a graph modeling problem [7]. Other studies, such as point-based taxi demand prediction [14], and station-based bike delivery strategy [31] are also formulated within topological data that are no longer Euclidean. Instead of CNN, graph convolution or even 3D convolutional networks [32] can effectively handle the non-Euclidean data.

The other line of studies applies the recurrent neural network to model temporal sequential dependencies [21]. However, these methods cannot effectively handle the observations that are periodic along a timeline or close in the space, but sometimes demonstrate very different behaviors. Recently, several studies attempt adopting the convolutional LSTM [33] to handle complex spatiotemporal prediction problems like taxi demand forecasting [34,35]. Though the convolutional LSTM (ConvLSTM) is able to settle these tasks, its training becomes difficult with the increase in network depth, which limits it to capturing wide-ranging spatiotemporal relationships. Yao et al. [36] propose a multi-view spatial-temporal network integrating LSTM, local-CNN, and semantic graph embedding [8]. This model introduces an attention mechanism to simulate periodic changes. Several studies have extended the traditional CNN and RNN structure to graph-based CNN and RNN such as graph convolutional GRU [6], multi-graph convolution [37], and gated attention on graphs [38]. Huang et al. [39] use the Stacked Autoencoder (SAE) and Deep Belief Networks (DBN) to predict the traffic flows. Chen et al. [40] construct a deep learning network, which uses time series functions to extract traffic flow characteristics. Koesdwiady et al. [41] adopt the DBN to predict traffic flows, in which the correlation between weather parameters and traffic flow is investigated. Chu et al. [42] propose the MultiConvLSTM, which can extract temporal and spatial characteristics from historical data and metadata to predict the origin–destination of traffic flows. Chu et al. [43] formulate the public transportation network as a graph, and develop a semi-supervised classification model based on Graph Convolution Network, which can directly estimate the waiting time of public transportation on the graph-structured data with limited labelled data.

Despite promising results on this single task, we argue that transition flows under the road map also play a guiding role in traffic prediction, thus it is possible to capture underlying factors by jointly optimizing this related task. For an instance, Wang et al. [44] propose a multi-task model that constructs a multi-task loss function to estimate the travel time of each local path and the entire path simultaneously, which can balance the tradeoff between individual estimation and collective estimation.

Different from the above-mentioned studies, we propose the attention-based deep spatial-temporal network with multi-task learning architecture (ADST-Net) for traffic flow prediction. The ADST can effectively represent the latent spatial similarity and temporal dependency. By modelling the transition flow forecasting as an auxiliary task, which potentially shares common factors with the main task, the flow prediction performance is boosted.

## 3. Materials and Method

In this section, we would like to establish some notations of the relevant concepts involved in our paper and define the metro flow problem. We adopt the similar definition as [12,24] and define the set of independent stations S = {S1, S2,…,Si} as location points of a city, and the past time series recorded by each station is defined as T = {I0, I1,…,It}. The prediction time interval is set to 10 min. For spatial data, more sophisticated partitioning methods can also be applied, such as partitioning the space with road network [14], or hexagonal partitioning [45]. However, this is not the focus of this paper. Furthermore, we regard the number of passenger check-in/check-out records during a fixed time interval for a specific station as the incoming/outgoing metro flow. Formally, y^tin and y^tout represent for the inflow/outflow value for station *i* during the *t*-th time frame. As a consequence, our ultimate goal is to predict:(1)y^t+1i, in&out = ℱ(Xt−h,…,t,εt−h,…,t)
where Xt−h,…,t are the historical flows (including station flows and transition flows) and εt−h,…,t are the external context features for all stations at each time slot from *t-h* to *t*, and y^t+1i, in&out is the future passenger flow of the stations on the whole metro network over the next time slice. We define our prediction function as ℱ(⋅) at all stations within the previous time intervals up to *t-h* to encode the complex spatial and temporal correlations. Briefly, our learning task ℱ:ℝT×|S|×2→ℝ|S|×2 aims to predict the metro flow for each station at time point t + 1 given the past recordings until time interval t. The mathematical notation utilized in this study is listed in Table 1.

We formalize the learning problem of spatiotemporal traffic flow forecasting as mentioned above. As deep learning methods are generally difficult to interpret, it is important to understand what contributes to the performance improvement. In this section, we focus on the issue of this specific task, and describe the details for our proposed ADST-Net with a multi-task learning framework, which is capable of learning the representation of traffic passenger flow from historical snapshots.

Figure 2 demonstrates the architecture of our proposed method. First, we briefly introduce the proposed ADST-Net with a multi-learning framework. Next, we present the approach to incorporate the attention mechanism and multi-scale convolution into the model. Finally, we show how to combine important spatiotemporal knowledge with gated external measurements into the supervised training procedure at each channel under a multi-task learning strategy.

### 3.1. Attention Based Deep Spatial-Temporal Network

To obtain spatial and temporal sequential dependency, combining local residual unit and external features through multiple channels have shown the state-of-the-art performance in crowd flow forecasting methodology. Here, the three channels utilized in our model are similar types of temporal segmentation used in [4], e.g., several time slots ago, one day ago, and a week ago. We further integrate an attention mechanism-based residual network and multi-scale CNN into the model to improve the prediction. Furthermore, we add an auxiliary task of transition flow under a similar architecture with main task to mutually reinforce the result of metro flow.

### 3.2. Fine-Grained Temporal Modelling

The purpose of attention is to select information that is relatively critical to the learning task from all input data. Regarding our temporal modeling, the motivation comes from channel-wise attention [46,47], which achieves reasonable results in the computer vision. This algorithm can automatically learn the weight of each channel, find the most correlation flow observations among different time intervals, and emphasize them by assigning higher weights. Through this weighted operation, the input channels are scaled according to the learned weight. In this work, considering that observations at the adjacent time frames own higher correlations, we adopt the relevant idea and generalize it for modeling fine-grained temporal properties across a sequence of observations within various channels.

Considering that the temporal relationship along the time axis is also critical for more accurate forecasting, we focus on explicitly quantifying the difference of the contributions in the time series. First, we suppose that the relative historical time intervals should be intercepted into different *P* temporal channels to handle the temporal dependency. For each channel, in order to deal with the fine-grained temporal correlations, we further select *Q* time slices for the different channels via a small grid search. For example, if we want to predict Tuesday 22 January 2019 at 9:00–9:10 a.m., we select 50 min before the predicted time (i.e., Tuesday 22 January 2019 at 8:10–9:00 a.m. and |*Q*| = 5), which is directly adjacent to the predicting time point, 30 min in the corresponding last day (i.e., Monday 21 January 2019 at 8:40–9:10 a.m. and |*Q*| = 3), and 20 min in the corresponding last week (i.e., Tuesday 15 January 2019 at 8:50–9:10 a.m. and |*Q*| = 2). These certain time frames *q* ∈ *Q* are used to handle the inherent temporal variation for each channel *p* ∈ *P*. The strategy behind these data segmentation come from different temporal frames’ influence and finally we decide these temporal intervals with a ratio of 5:3:2 through a small grid search as illustrated in Section 5.

Furthermore, we utilize a rectified block as an attention mechanism whose architecture is given in Figure 3. Here we use |*Q*| = 5 for the “recent” channel as an example (actually all-time slices of different *P* temporal channels are fed into the model together), suppose that temporal observations *q* ∈ *Q* and a single snapshot Xt∈ℝ|S|×C denote the *t*-th reading where *S* refers to all stations and *C* is the feature dimensions. Note that *C* would be two if the feature only contains incoming and outgoing flows. Thus, the pipeline of attention mechanism could be defined as follows:(2)zt = Fpool(Xtc1) = 1|S|∑i = 1|S|Xt, i,: for t = 1,2,…,Q
(3)rt = Fratio(zt) = ReLU(W1(zt/ratio)+b1)
(4)st = Fatt(rt) = σ(W2rt+b2)
where we leverage the global average pooling Fpool among all stations Xtc1 (c1 denotes a specific channel) to produce a summary of each time frame into a scaler zt. After that, an attention operation Fatt with a squeeze function Fratio is conducted to generate adaptive weights st by applying non-linear transformations on zt, where σ is the sigmoid function, respectively. The significance of ratio is that the model could fuse the spatiotemporal information into a lower dimension. Both the Fratio and Fatt are simply dense operations but own different active functions. ReLU (Rectified Linear Unit) function is utilized inside Fratio.

Formally, the important measurement $s^{t}$ derived by the attention mechanism across all stations is applied to scale each temporal recording.
(5)$$X\prime_{t}^{c_{1}}\;=\;s_{t}⊙X_{t}^{c_{1}}$$
where ⊙ is the Hadamard product (i.e., element-wise multiplication) of two vectors and $X\prime_{t}$ measures the importance of each certain time frame *q* in the current channel *p* ∈ *P*. We finally capture the fine-grained temporal properties and obtain the re-weighted representation of each channel.

Followed by rectified block, we append several similar blocks in terms of the property within a residual network that consists of Batch Normalization (BN), Rectified Linear Unit (ReLU), and Convolution (Conv). Then, a temporal module with residual connections is built which could be repeated several times, as illustrated in Figure 4

### 3.3. Latent Spatial Similarity

The Standard CNN is used to characterize the spatial interactions. As suggested in [8], it is hard to achieve distant spatial information through a traditional CNN, and then generalize it to cases where functional similar stations with strong correlations might not be close in space. Additionally, using stations with weak correlations to predict a target station actually hurts the performance. Inspired by the work in [48], we propose a variant CNN module. Thus, we use a novel multi-scale CNN module to enforce explicit spatial similarity with respect to the size of surrounding neighborhood map. The multi-scale ConvLSTM in [48] uses different levels of resolutions, while we use different kernel sizes.

For each time interval *t*, instead of treating the spatial data as a standard 2D grid map or topological graph, we treat the target station *i* and other citywide stations as Xt∈ℝ|S|×2, where S refers to the station set and 2 represents the incoming and outgoing flows of a station, then adopt one-dimensional convolution to express spatial appearances. The reason why we assemble the inflow and outflow of multiple stations at different times into an image-like matrix for the convolution network is that, two kinds of flow variables with practical physical significance could be regarded as two features (like RGB patterns for images) and can be used to extract latent correlations among the given stations with convolutional network.

As shown in Figure 5, the multi-scale CNN utilizes three filters with a different convolution kernel size ξ (which refers to the power of coverage scope for stations) on this input cube, then pools the measurements to fuse different aspects through metro graph transformations. The spatial feature extraction would focus on close stations with a small reception field, i.e., neighbors that could be reached through a small number of steps, while local flow from other stations in larger areas might not be fully covered. Adding a multi-scale convolution module would increase the reception field and consequently lead the model to capture more latent relationships. For a single CNN, it takes $X_{t}$ as input $X_{t}^{\left(0\right)}\;$, and the formulation of each convolutional layer *k* is:(6)$$y_{t}^{\left(k\right)}\;=\;ReLU\left(W^{\left(k\right)}X_{t}^{\left(k-1\right)}+b^{\left(k\right)}\right)$$
where $W^{\left(k\right)}$ and $b^{\left(k\right)}$ are the learning parameters, and we use ReLU as the activation function. After stacking the *K* convolutional layers in each scale, three measurements are further merged based on a parameter matrix to infer the spatial correlation for each channel, and output the predictive result as $X_{main}$ to the multi-task fusion strategy. In our work, the final mergence in this multi-scale convolutional strategy is conducted with a simple element-wise summation operation.

Using the aggregation of multi-scale convolution, the proposed formulation ensures that information coming from stations that are distant in space, but very close in terms of functionality (such as two central business districts), could be correctly exploited.

### 3.4. External Information

In order to deal with the insufficiency of the input domain, we finally preserve the external features (e.g., temporal segment and functional stations) by calculating the summarized global information or local measurements. There are many external factors that may influence the flow prediction. However, among them, the rush hours and weather conditions play the main role in flow prediction [49]. Considering that a sudden increase/decrease in flow usually indicates the occurrence of extreme weather or accidents that would affect the future movement of people across stations and block the traffic, we obtain many kinds of descriptive information of a particular station or a time frame, denoted as εtS∈ℝext, and these vectors could be incorporated with spatiotemporal properties through a feature embedding operation or gated mechanism. For example, rush hours could be mapped to the collection {0, 1, 2} with an embedding layer, which is a common discretization method to make nominal variables that mark 0-peak, 1-plain, and 2-valley with a numeric identifier, and weathers could also be marked as 0-rainy and 1-sunny. This would be further reshaped into the same dimension with the station flow feature matrix by means of a fully connected layer. The fusion strategy for two parts is formulated as follows:(7)Yt+1 = ReLU(Wt⊙Xt+Wext⊙εtS)
where Wt and Wext are the weights for each measurement. It is noteworthy that the transition flow matrix Et adopts the same method in Equation (6) to obtain the future transfer volume. Since the proposed framework is flexible in incorporating additional features, there are several external features that we decide to consider for the predictive model: hour of a day, day of a week, weather conditions, peak identifiers and sluice machine quantity. Such external properties may have varying degrees of impact on metro flow, as depicted in Figure 6.

As can be observed in Figure 6a, the flows on weekdays demonstrate a significant difference from those on holidays and weekends in most stations due to people’s daily routine of work, while the flows on holidays increase randomly since the coming of the Spring Festival. For similar reasons, in Figure 6b, railway stations demonstrate the largest flows and the business region is predictable due to the users’ regular travel patterns (e.g., from office to home). In Figure 6c, there is also a difference between rainy and sunny days for the total traffic flow, which could be encoded as a predictor. From Figure 6d, we could infer that stations with a larger number of sluice machines exhibit higher throughputs and this attribute would be regarded as one numerical feature.

### 3.5. Multi-Task Learning

Specifically, a multi-task learning framework is adopted to jointly learn the main task, i.e., predicting the traffic flow at each station as well as an auxiliary task ℱtrans:ℝT×|S|×|S|→ℝ|S|×|S|, i.e., to predict the transition flow between each two stations [50], as we assume that the single station’s flow and the transition flow among stations mutually influence each other, and would further reinforce our target prediction by factors that are shared with this relevant auxiliary task.

We denote Et∈ℝ|S|×|S| as the historical transition flow that could be obtained by exploring the user’s identifier and extracting the transfer volume from one station to another under each time interval according to the destination station’s historical swiping record. The subsequent methodology for Et is similar to the main task and finally outputs the predicted transition flow denoted Xaux.

It is noteworthy that the representations learned from our main task and auxiliary task should be connected given the strong correlation between them. Considering that our two measurements, $X_{t}$ and $E_{t}$, mean two different flow prediction tasks and their own different shapes, we choose concatenation instead of sum aggregation to fuse the two latent representation maps of $X_{main}$ and $X_{aux}$, which could better integrate two tasks by mutually reinforcing them. As we could see in Figure 7, after this fusion strategy, we append a convolutional layer into the fused measurement, which is used to transform the combined latent feature tensor into individual outputs of our two tasks. At the end of our network, the external information $\varepsilon_{t}^{S}$ is further incorporated with each task, respectively, and we finally obtain the flow prediction results.

According to the above figure, here we give our final forecasting function, which is defined as:(8)[Yt+1, ℋt+1] = ReLU(Wfℳ(Xt,Et, εtS)+bf)
where Wf and bf are the learned parameters that adjust the degrees affected by those domain knowledges, while the ℳ(⋅) function defines a series of operations including the “3C-ARM” component, concatenation and decomposition module, and the external features incorporation. Yt+1∈ℝ|S|×2 owns two vectors, y^t+1i,in and y^t+1i,out, which define the incoming/outgoing flow for each station mentioned in Equation (1). Note that in this paper, we focus on the main task and only evaluate Yt+1 in all of the following experiments. The output of our model is {0, 1} since we normalize the value of flow with Min–Max normalization instead of random guessing normalization. Later we re-scale the predictive result Yt+1 back to the normal values and compare it with ground truth.

The novelty with respect to those previous approaches under the idea of multitask learning is that, our designed architecture indicates a hard parameter sharing framework where two tasks are connected by coupling their latent representations of the middle layers, then trained together to share the key component of the model, and finally own a dedicated output layer for these two tasks.

The thought behind this strategy is that different stations pose varying impacts on a specific station over past time slices. Our proposed related task could be explained using a shared set of factors, and the multi-task learning framework would take full advantage of the training information in this auxiliary task to improve the generalization and avoid overfitting.

### 3.6. Loss Function

Our ADST-Net is trained by a loss function. In this section, we provide details about the loss function used to formalize this learning problem and jointly train the proposed model. The loss function for the main task is defined as:(9)$$\;{ℒ}_{main}\left(\vartheta_{1}\right)\;=\;\frac{1}{S}\overset{s}{\underset{i}{\sum }}\left(\left|{\hat{y}}_{i,t}^{in\&out}-y_{i,t}^{in\&out}\right|+\beta_{1}\left(\frac{{\hat{y}}_{i,t}^{in\&out}-y_{i,t}^{in\&out}}{y_{i,t}^{in\&out}}\right){}^{2}\right)$$

The loss for the auxiliary task shows a similar format:(10)$$\;{ℒ}_{aux}\left(\vartheta_{2}\right)\;=\;\frac{1}{S}\overset{s}{\underset{i}{\sum }}\overset{s}{\underset{j}{\sum }}\left(\left|{\hat{y}}_{i,j,t}^{trans}-y_{i,j,t}^{trans}\right|+\beta_{2}\left(\frac{{\hat{y}}_{i,j,t}^{trans}-y_{i,j,t}^{trans}}{y_{i,j,t}^{trans}}\right){}^{2}\right)\;$$
where $\vartheta_{1}$ and $\vartheta_{2}$ are the learnable parameters in the ADST neural network, and the loss function mainly adopts the format of mean absolute error. Furthermore, the mean absolute error is more sensitive to predictions of large values. To avoid the loss being dominated by large value samples, we introduce a formula with a proper weight decay $\beta_{1}$ or $\beta_{2}$ as regularization, which could smooth the large prediction differences. In our experiment, we use a weighted sum combination for these two loss functions as the final learning objective of our model with corresponding weights as the hyper parameters $\varphi_{1}$ and $\varphi_{2}$. The individual loss function is defined as follows:(11)$${ℒ}_{ADST}\left({\hat{y}}^{\left(t\right)},y^{\left(t\right)}\right)\;={\;\varphi }_{1}{ℒ}_{main}+\varphi_{2}{ℒ}_{aux}$$

In brief, the algorithmic procedure for our learning method is outlined in Algorithm 1.
**Algorithm 1.** Training Pipeline of ADST-Net Algorithm. The training pseudo code of the ADST-Net algorithm. During each iteration, we optimize the objective (Equation (11)) on the selected batch of training samples. **Input:** historical observations: {Xt,Et|t=t1,t2, …,tT};
   external information: {εt|t=t1,t2, …,tT};   target variables: yt∈ℝ|S|×2
**Output:** ADST-Net Model*// build training* samples1  Initialization Strain←∅
2  ***for***
t∈T
***do*** // T denote the intercepted time series segments of closeness, period and trend from available observation set.3     append a training instance ($\{X_{t},E_{t},\varepsilon_{t}\},y_{t}$) into $S_{train}$
4  ***end****5  // train the model*6  Initialize all learnable parameters $\vartheta_{1},\vartheta_{2},\omega_{f},b_{f}$ in ADST-Net7  ***repeat***8     randomly select a batch of instances $S_{batch}$ from $S_{train}$
9     optimize $\vartheta_{1},\vartheta_{2},\omega_{f},b_{f}$ by minimizing the loss function (11) with $S_{batch}$
10  ***until*** stopping criteria is met11  Output the learned ADST-Net Model
where $X_{t},E_{t},\varepsilon_{t}$ denote the input of our ADST model, and $S_{train}$ is the training set. First, the training samples are constructed from the original data. After that, forward propagation and back propagation are repeatedly applied to train the model. The Adam algorithm is adopted for optimization.

## 4. Experimental Results and Analysis

In this section, we demonstrate our experimental results in detail. In order to make our work more comprehensive and fully convincing, we consider validating the model using two different datasets and compare our proposed model with other state-of-the-art baselines, including classical time-series model, LSTM-based and spatiotemporal-based models for citywide-level traffic flow forecasting.

### 4.1. Dataset Description

We evaluate our proposed method on two citywide public datasets from Hangzhou and Beijing City.

MetroHZ: An open dataset from Ali Tianchi competition (https://tianchi.aliyun.com/competition/entrance/231708/information). The raw data information of MetroHZ is as follows:{Time, lineID, stationID, deviceID, status, userID, PayType}Time: The timestamp of each metro card swiping record.LineID: The metro line {A, B, C}.StationID: The stations {0–80}.DeviceID: The sluice machines of each station.Status: User’s check-in and check-out records {0, 1}.UserID: The user’s identifier.PayType: The type of user’s swiping mode.

This adequate dataset contains approximately 70 million records of Hangzhou, from 1 January 2019 to 25 January 2019. In the experiment, we use data from 1 January 2019 to 24 January 2019 (totaling 17 regular working days) as the training data, and the remaining day (25 January 2019, 19 January 2019, 20 January 2019) and an additional day (28 January 2019) as the test set under the consideration that only a single day for testing purposes is indeed insufficient. Such a test setup is also able to measure the generalization and transfer ability.

There are 81 stations in total within three lines in Hangzhou and the traffic flow relating to a certain station in a time interval falls into two attributes, i.e., incoming and outgoing flow. The flow matrices for the stations in a series of time slices could be constructed with “Time”, “StationID”, and “Status” from the raw data through aggregation. Hence, each element in our calculated flow matrix denotes the total check-in/check-out value of the corresponding stations in a time frame, and we finally obtain those metro condition snapshots that comprehensively reflect the metro flow of all stations with original information. For the external information $\varepsilon_{t}$, we totally extract five variables including weeks, hours, weather, sluice machines and peak identifiers, then map these fine-grained attributes into a specific vector with 20 dimensions. The dataset used in this paper are described in Table 2.

TaxiBJ: An additional public dataset is TaxiBJ (https://www.microsoft.com/en-us/research/publication/deep-spatio-temporal-residual-networks-for-citywide-crowd-flows-prediction), which includes taxicab GPS data and meteorology data in Beijing from four-time zones. We select TaxiBJ16 (1 November 2015–10 April 2016) as our second dataset in this paper.

In accordance with [4], we obtain two types of crowd flows. In the experiment, the data from 1 March 2016–31 March 2016 is used for training (31 days), and the data consisting of 2 April 2016, 3 April 2016, 4 April 2016, 5 April 2016 is chosen for testing. Besides, the time interval is split by 30 min and the grid map sizes are constructed with (32, 32).

### 4.2. Preprocessing

After the construction step of the flow matrix described above, we group together all observations from those points, and sort them in an ascending way based on their past time series, while the input of the network consists of 10 historical observations including five recent components, three daily components, and two weekly components. The reason why we intercept the historical time slices into three channels is that “recent” means the recent past traffic passenger flows that inevitably have great influence on future flows, while “daily” is used to cover the regular daily pattern and “weekly” is used to capture the weekly attributes in traffic data. The motivation behind the decision for using these intervals is that the |*Q*| values (5, 3, and 2) for the different channels are chosen via a small grid search ahead of evaluation on MetroHZ, as we could see in Table 3.

We use Min–Max normalization to convert metro flow values for all stations to the {0, 1} scale. Meanwhile, we adopt an embedding layer to encode the external properties (e.g., holidays, weekdays, and weather conditions which are collected from the historical weather report). After prediction, we apply the inverse of the Min–Max transformation obtained by the training set to recover the metro flow value and use it for the evaluation. We complete the sample generation step on the training and testing set after the preprocessing phase.

For MetroHZ, when finishing our prediction, we store the results and post-process the data by filtering the samples with flow values greater than 3000 or less than 2 for some specific stations in Hangzhou, which is a common operation used in simulation experiments to remove outliers [8]. In some stations, it would never reach the upper bound like 3000 according to the time interval and metro establishment. It is unnecessary to care about such traffic scenarios with irregular metro flow that would affect the overall performance in real-world applications.

### 4.3. Experimental Setup

For MetroHZ, we select 17 days as the training data to learn the models, and the remaining 4 days is chosen as the testing set. As for TaxiBJ, we chose 31 days for training and 4 days for testing. When testing the result, we use the previous 10 time intervals to predict the traffic flow in the next time interval. The Mean Absolute Error (MAE), Mean Absolute Percentage Error (MAPE) and Median Absolute Error (MdAE) are used to evaluate our learning algorithm, which could be defined as follows:(12)$$\mathrm{MAE}\;=\;\frac{1}{N}\overset{N}{\underset{i\;=\;1}{\sum }}\left(\left|{\hat{y}}_{t+1}-y_{t+1}\right|\right)$$
(13)$$\mathrm{MAPE}\;=\;\frac{1}{N}\overset{N}{\underset{i\;=\;1}{\sum }}\frac{\left|{\hat{y}}_{t+1}-y_{t+1}\right|}{y_{t+1}}$$
(14)$$\mathrm{MdAE}\;=\;\mathrm{median}\left(\left|{\hat{y}}_{t+1}-y_{t+1}\right|\right)$$
where ${\hat{y}}_{t+1}$ and $y_{t+1}$ denote the prediction and ground truth of station *i* for the time frame t+1 , respectively, and *N* is the total number of training samples. Instead of N-fold cross validation, we take rolling-origin-recalibration evaluation [51] into account and conduct our validation based on forward-chaining method, which is a common strategy of nested cross validation for time-series data. In detail, the MetroHZ utilizes three splits because we need to ensure that at least one day of training and validation data is available. Thus, 07/01/2019, 18/01/2019 and 25/01/2019 are tested separately, and we finally achieve an average MAE of 15.20, which would reflect a robust estimation of the model error.

We adopt ReLU as the activation in most of the convolutional networks, and the employed neural network has a number of output neurons equal to the number of metro stations in last layer at the top of the model, while the hidden layers inside the neural network own their neurons according to the designed architecture. All the models are implemented using Keras, which is based on TensorFlow as its backend engine, and optimized via Adam to perform all updated weights. Conventional network parameters are initialized using a uniform distribution with the default parameter. The batch size in our experiment is set to 64; afterward, we train our model on the full training data for 100 epochs. The learning rate is initially set as *e* × 10^−3^ and decreases by half within 10 runs. We also use early-stopping (whose round was set to 10) on the validation set in all experiments by monitoring the best score, which can choose the best generalizing network during training. In the multi-task learning, the weighted linear function is used to aggregate the loss function from the proposed two tasks, while the final ratio is set as 5:1. Besides, we evaluate different hyperparameters for them all, finding the optimal parameter setting for each according to MAE.

In order to study the effects of different hyper parameters of the proposed model, we evaluate models on our dataset by varying three of the most important hyper parameters in Figure 8, i.e., the stacked residual connection layers, the $l_{ratio}$ in rectified block, and the reception field size ξ for multi-scale operation.

The dimension of each hidden representation of deep residual network is 64 and the total number of hidden layers is 160, because there are 20 residual units in our model through the tuning in Figure 8a. As the network goes deeper, the MAE first decreases and then increases since there is a much more difficult training process. When the residual connection layers reach 20, our proposed model achieves the best performance.

For temporal information, we intercept the length of past time interval for each channel as recent |*Q*| = 5, daily |*Q*| = 3 and weekly |*Q*| = 2, respectively. The hyper parameter called $l_{ratio}$ in rectified block suggests the fine-grained segmentation of temporal properties, and we also find that the ratio inside the rectified block affects the performance from Figure 8b. We finally select the suitable property because the loss slightly degrades but mainly remains stable after the value of 8.

For spatial information, we set multiple convolution kernel sizes ξ to 3 × 3, 5 × 5 and 7 × 7, where the super parameter ξ is determined by trials. The adjacency matrix is adopted to define the neighborhood and retain the spatial information of the original metro network. Notice that the balance of the hyper parameter ξ refers to the power of coverage scope for stations and it will be further fine-tuned. Figure 8c shows the prediction MAE error with respect to this size, where the error first decreases and then plateaus with the increase in ξ. Larger ξ will enable the model to cover a larger region, which helps capture more station-wise correlation at the cost of increased model complexity and being more prone to overfitting. We choose the reception field size when the model performs best and remains consistent.

After optimization, there are four repetitive residual connection layers. As the network goes deeper, the MAE increases since it is much more difficult to complete the training process. For other hyper parameters inside the spatiotemporal component, we finally select the suitable property as the loss is slightly degraded and remained mainly stable.

Our simulation is programmed in Python 3.5 and run on Ubuntu 16.04 with the CPU Intel i5-4440 and 16 GB RAM. The whole training process takes about 25 min.

### 4.4. Baseline Comparison

To fully evaluate our proposed algorithm, a series of recent state-of-the-art approaches for spatiotemporal traffic flow forecasting are also implemented and used as competitors.

ARIMA: Classical approaches coming from spatiotemporal time-series models [10]. Here we utilized ARIMA with 10 time lags to generate our prediction results.STARIMA: A spatial-temporal ARIMA model integrating spatial and temporal information to predict traffic flow [12].LSTM: Containing 512 hidden units with 1-time steps [22]. The result show that the spatiotemporal information is indeed useful compared with the single LSTMs.GeoMan (Liang et al., 2018): A multi-level attention-based recurrent neural network for predicting the readings of a geo-sensor over several future hours [52]. It consists of a multi-level attention mechanism, and a general fusion module to incorporate the external factors from different domains. Five time slots of historical traffic flow are utilized to predict one time slot of future traffic flow.STNN (Delasalles et al., 2019): A dynamical spatiotemporal model [53] using a recurrent neural network to model time series of spatial processes. The input of this network contains an information matrix of points and an adjacency matrix. The key parameter configurations (latent space N = 5, sparsity regulation γ = 0.01 and soft-constraint λ = 0.01) are fine-tuned.DMVST-Net (Yao et al., 2018): A multi-view based deep learning approach [8] for traffic demand prediction. It includes three different views: temporal view, spatial view, and semantic view, which are modeled by local CNN, LSTM, and graph embedding, respectively.ST-ResNet (Zheng et al., 2017): A deep learning-based framework [4] for traffic prediction, which constructs a city’s traffic density map at different time slices as images. A CNN with residual units is used to extract features from historical images and model citywide spatial dependency.

Here we summarize the key components for these baselines and our methodology in Table 4:

As shown in Table 4, ARIMA only focuses on temporal modeling. STARIMA introduces the spatial information besides the temporal information. GeoMan and DMVST-Net cannot apply to multiple channels. The STNN does not consider the external factors. ST-ResNet rarely considers other meaningful tasks to improve the model.

To comprehensively evaluate these approaches, three metrics including MAE, MAPE and MDAE are used. We present the performance on the testing dataset over several runs. The errors are calculated between our estimator and ground truth.

The experimental results are shown in Table 5. From the table, we can see that our proposed ADST-Net achieves the best performance regarding MAE against other comparative methods, and generally demonstrates consistent improvement of more than 38.29 MAE points across different days on two datasets, which is calculated according to the average difference between ADST and others on four days.

All the prediction results are recorded in the table referred to at the following link https://docs.qq.com/sheet/DUVliRmpjWVdqSmdh.

Compared with these state-of-the-art baselines, our model turns out to be more accurate for traffic flow predictions on all these testing days, including ordinary weekdays, weekends and special holidays.

Here, we display the performance of our proposed method compared with other competing methods on a specific day. Lower MAE denotes better prediction accuracy, as well as the other two metrics. For MetroHZ dataset, the proposed ADST-Net significantly outperforms all competing baselines by an obvious margin. As for the other dataset TaxiBJ, the model also owns an adapted architecture, which yields a higher prediction accuracy using reasonable training time, as the following Table 5 shows:

Table 5 renders that (1) classical time-series approach (like ARIMA) aims to discover temporal patterns, but loses latent spatial dependency. (2) As a variant of ARIMA, STARIMA applies for the stable series prediction and cannot accurately forecast the series with sudden and dramatic change. (3) Although spatiotemporal LSTM provides promising results, the LSTM operation consumes more time due to its complex network. (4) Other previous models leverage different kinds of spatial-temporal convolution or RNN-based architectures. However, these approaches mainly focus on neighbors, but neglect the multiple reception field of spatial similarity and fine-grained temporal measurements, which would result in limited contributions for model training. (5) Besides, ADST-Net performs at an appropriate training speed. Although some methods spend less time on the training phase due to the simple network architecture, they reveal a higher prediction error.

We can see that the results for MetroHZ reflect a substantial decay in prediction error on the basis of regular weekdays and weekends; how would our model perform in holidays, such as New Year’s Eve or other major bank holidays? We continue our prediction for TaxiBJ on four days, including 4 April 2016, Tomb-sweeping Day in China, which could evaluate our model’s performance during holiday periods.

Furthermore, we discover that predictions on weekends are slightly worse than the results on weekdays under some methods. On one hand, the residential stations might have a higher flow during the morning peaks of the weekdays, as people need go to work. Such regular daily routines are less likely to occur on the weekends. Therefore, the forecasting task is much harder for the weekends. On the other hand, the dataset contains a few samples for training, however, our model still renders more robust predictions in general.

In the meantime, the total trainable parameters amount to 691,799 under the corresponding training and testing time.

Our results demonstrate the effectiveness of the attention mechanism and multi-scale convolution strategy in characterizing fine-grained spatial-temporal correlations. Moreover, the multi-task learning framework offers a clearly better performance due to the common factors shared from the proposed transition volume prediction task within the same network architecture.

Except for the major learning function ℱ:ℝT×|S|×2→ℝ|S|×2 for our forecasting, we also conduct an extended experiment with ℱ′:ℝT×|S|×2→ℝU×|S|×2 which represents our prediction at time steps *U* in the future based on MetroHZ. As shown in Table 6, we feed the model’s output as its input for multiple steps ahead prediction and our ADST-Net also shows better performance.

### 4.5. Effect of Model Components

To further explore the effect of spatial appearances and temporal dependency modeling, we also carry out the following evaluation of ADST-Net by removing different components from the model including (1) the rectified block-based attention mechanism for temporal modeling, (2) the multi-scale convolution for explicit spatial appearances, (3) an auxiliary task of transition flow prediction, (4) external information, and (5) the other two channels that represent the daily and weekly. The results are rendered in Table 7, which show that the removal of any component in our model would cause an error increase.

From Table 7, we could further see that each component confirms its effectiveness for metro flow accurate forecasting.

Attention mechanism: for this variant, we only focus on spatial modeling while dropping the temporal attention mechanism by removing the rectified block. The result is worse performace given the meaningful role of the temporal correlations.Multi-scale convolution: without this method, our model is unable to cover most of the stations in the city based on only a single reception field, and the performance becomes worse.Auxiliary task of transition flow: this auxiliary task considers the potential factors that could be shared from the transition volume. Intuitively, the predictive results would provide valuable information as the hidden stream between unconnected stations and greatly influence the flow of a single station. The experimental results show that by taking the auxiliary task into account, we decline the MAE from 13.85 down to 13.28.External information: in this variant, we extended influencing factors that include not only the traditional weather condition and week identifier, but also new attributes such as the sluice machine quantity and functional area flag. Our model performance degrades without using the external information.Multiple temporal channels: it is necessary to intercept history observations into three parts that denote recent, daily, and weekly to load our model with a clear view to handle dynamic complex spatiotemporal interactions.

### 4.6. Result Analysis

In this paper, we focus on the metro flow of key time periods on workdays, which could differ from Monday to Friday. Additionally, different stations would have potential correlations in the same time interval. Based on this insight, we visualize forecasting results with ground truth to better understand the model. The results show that our forecasting algorithm performs slightly better in general in tracing the ground truth curve of the metro flow in most of the key time segments at all kinds of relevant stations, owing to our explicable network.

Hence, our model could achieve the best forecasting performance, which would be helpful in policy making for subway departments. We believe that incorporating our attention mechanism, multi-scale convolution, and auxiliary task strategy to existing methodologies would result in a statistically significant error decrease. All the codes for the above experiments are released at https://github.com/wumingyao/ADST.

Compared with the single-step prediction errors shown in Table 5, Table 6 indicates that the multistep prediction may result in larger errors (i.e., a larger MAE, MAPE, and MDAE) due to error accumulation. As the number of prediction time step increases, the single-step prediction errors accumulate. Each predicted value is fed back as a new input. As the prediction window moves on, the actual data in the window gradually reduce, which results in higher prediction errors. To reduce the accumulation errors is the main task for the multistep prediction.

## 5. Conclusions and Future Work

Traffic flow is a fundamental metric reflecting the state of the metro condition. It is incredibly challenging to forecast due to the high nonlinearities and complex patterns. To address this key issue, in this paper, we investigate the citywide-level traffic flow forecasting problem and propose the ADST-Net, i.e., an attention-based deep spatial-temporal residual network with multi-task learning to predict urban traffic flow. To extract the spatial–temporal correlations, our model uses three independent channels to extract the recent, daily-periodic and weekly-periodic fine-grained temporal correlations, respectively. Each channel combines a gated attention mechanism for fine-grained temporal modeling and multi-scale CNNs for latent spatial similarity mining into the residual network. Our proposed model cannot only capture the daily flow trends, but also can represent the sudden flow changes (e.g., Spring Festival) of all stations. The attention mechanism emphasizes the critical temporal periods with higher weight, which improves the flow prediction accuracy. The flow prediction also benefits from the usage of external embedding including the weather condition and rush hours. Furthermore, our proposed algorithm leverages the multi-task learning to improve flow prediction accuracy by modelling neighboring station flow as the auxiliary task. The experimental results on two real-world datasets show that our ADST-Net could achieve significantly better results against existing models. For further study, it is suggested that rich road network information should be considered with an appropriate fusion mechanism.

Several potential extensions in this research are considered. For example, the long-term prediction performance of our proposed algorithm is degraded due to the error cumulation. As our future work, directly modelling the relationships between the historical and future time steps may alleviate the effect of error propagation to improve prediction accuracy. Meanwhile, considering that the flow volume may fluctuate irregularly and suddenly (e.g., because of an occasional accident), the correlations between different time steps may demonstrate a completely different pattern from the past time. Therefore, the impact of uncertainty (e.g., unexpected things happen, and noisy observations) on the prediction accuracy of the model should also be considered in the future.

## Figures and Tables

**Figure 1 sensors-20-04574-f001:**
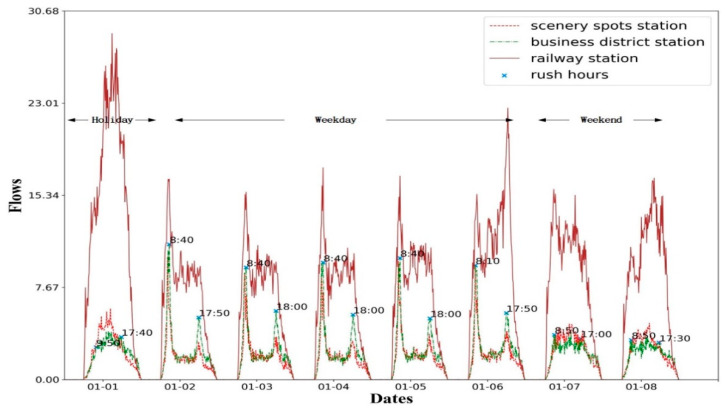
In/Out flow distribution at three different functional stations during the holidays, weekdays, and weekends, where each day consists of 144-time slices with the fixed time interval set as 10 min.

**Figure 2 sensors-20-04574-f002:**
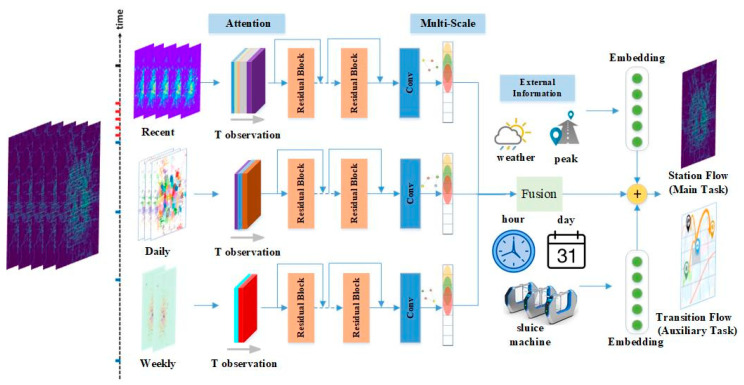
The system architecture of our proposed attention-based deep spatial-temporal network with multi-task learning architecture (ADST-Net) for metro flow prediction. The historical spatiotemporal observations are intercepted into three independent channels which share the same network structure, then three groups of data constructed by different snapshots are fed into a deep residual network with attention mechanism and multi-scale strategy, external information are fused at the top of the model, finally our ADST-Net is jointly trained on the target station flow (main task) together with a transition flow task (auxiliary task) using supervised signals under a multi-task framework.

**Figure 3 sensors-20-04574-f003:**
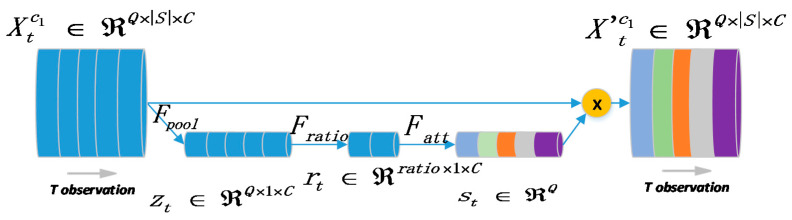
The rectified block as the attention mechanism to provide the fine-grained weights for the original observations in each channel.

**Figure 4 sensors-20-04574-f004:**
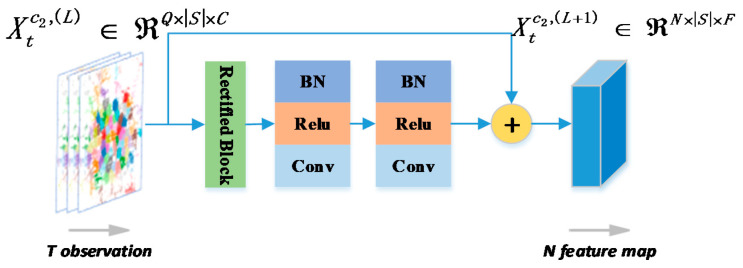
The fine-grained temporal correlation modelled by stacking a rectified block and a Batch Normalization (BN)–Rectified Linear Unit (ReLU)–Convolution (Conv) (BRC) block within one residual connection. Note that there are several residual connections in our model.

**Figure 5 sensors-20-04574-f005:**
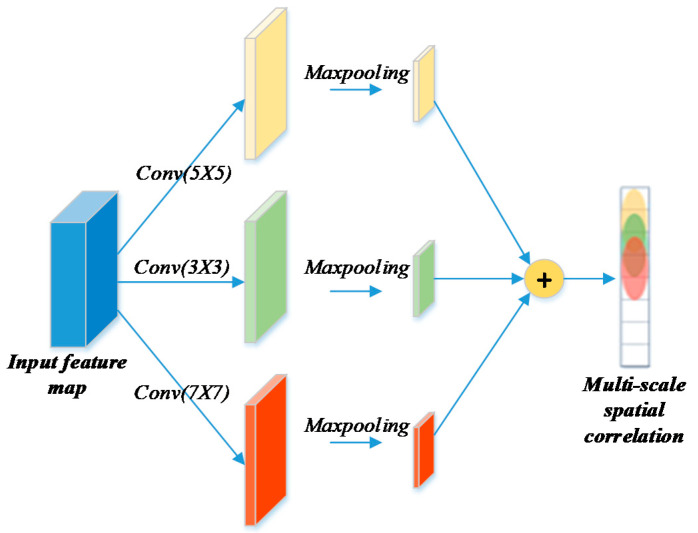
The proposed multi-scale convolutional strategy possesses higher adaptability for a diversified reception field.

**Figure 6 sensors-20-04574-f006:**
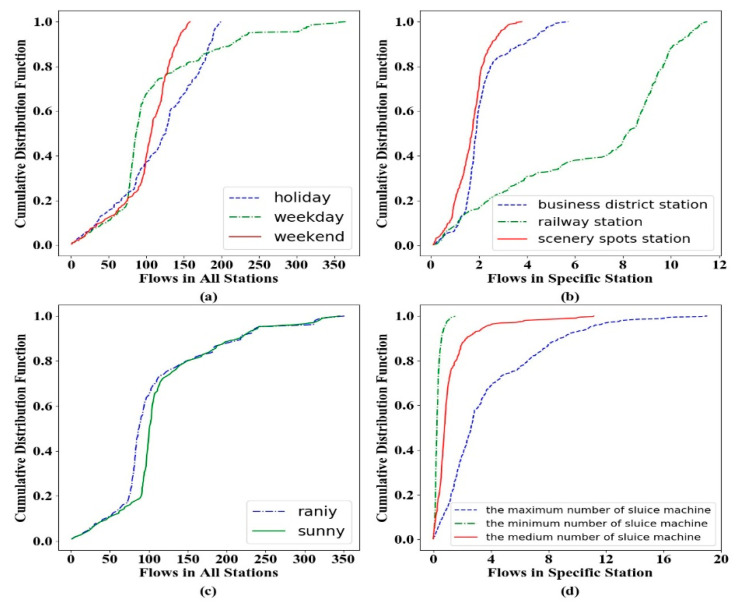
The CDF (cumulative distribution function) curve with solid and dotted lines represents the difference in traffic flow under various specific conditions, illustrating the effectiveness of several potential features that will be incorporated as external information. (**a**). Flows comparison on weekdays, holidays and weekends. (**b**). Flows comparison on business district station, railway station and scenery spots station. (**c**). Flows comparison between rainy and sunny. (**d**). Flows comparison among different numbers of sluice machine.

**Figure 7 sensors-20-04574-f007:**
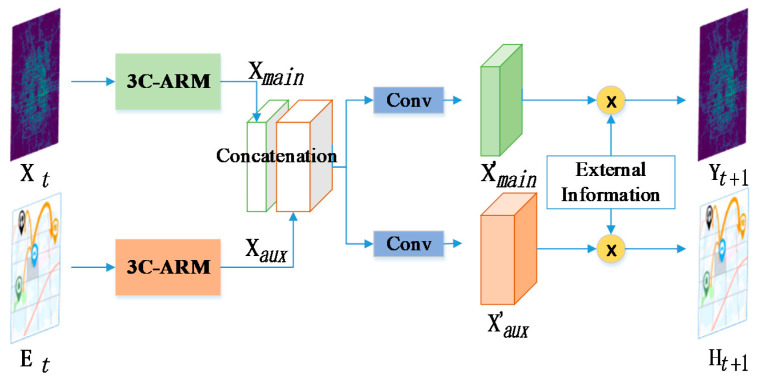
Our fusion strategy under the multi-task framework. The “3C-ARM” component represents the Attention mechanism, Residual connection and Multi-scale convolution within three temporal Channels. The output of “3C-ARM” is denoted as $X_{main}$ and $X_{aux}$, then we fuse them in a concatenated way, after which a convolution layer is appended to obtain the respective measurements $X\prime_{main}$ and $X\prime_{aux}$ of the two tasks. Finally, we feed them with our proposed external information $\varepsilon_{t}^{S}$ to a fully connected layer, respectively, to obtain the final results $Y_{t+1}$ and ${ℋ}_{t+1}$.

**Figure 8 sensors-20-04574-f008:**
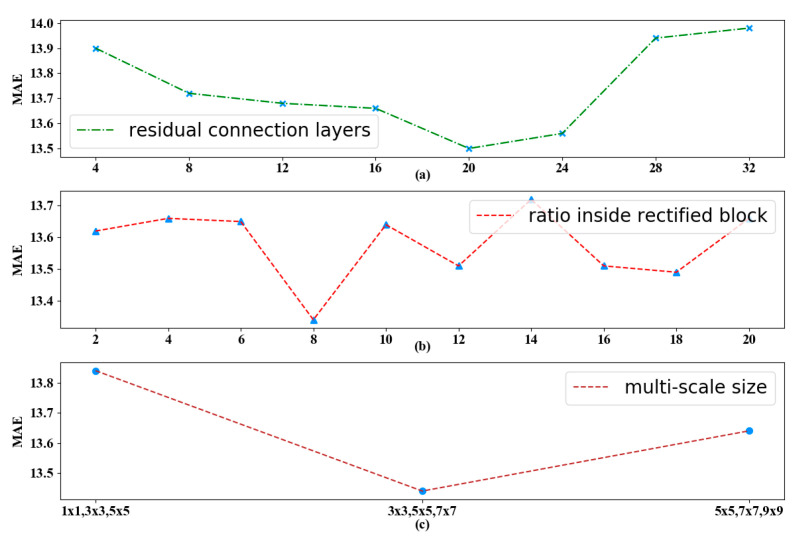
The tuning process for the key hyper-parameters inside our model. In detail, the ratio stands for temporal modeling and reception field size suggests reasonable spatial dependencies, while the residual connection layers control the total depth for our network. (**a**). The influence of different residual connection layers on MAE. (**b**). The influence of different ratios inside rectified block on MAE. (**c**). The influence of different multi-scale sizes on MAE.

**Table 1 sensors-20-04574-t001:** Notation description in this paper.

Notation	Description
S	number of metro stations
T	available past time frame set
Xt∈ℝ|S|×2	the historical incoming and outgoing flow observations for all the |S| stations at time t
Et∈ℝ|S|×|S|	the snapshot of historical transition flow among stations at time t
εt∈ℝext	external influencing factors
y^t+1i,in&out∈ℝ|S|×2	final prediction flow for all stations at future time t + 1

**Table 2 sensors-20-04574-t002:** Dataset Statistics.

Dataset	MetroHZ	TaxiBJ
Location	Hangzhou	Beijing
Time Span	1 January 2019–25 January 2019	1 March 2016–31 March 2016
Time interval	10 min	30 min
# Available Time frames	3600	1488
Shape of $X_{t}$	81 × 2	32 × 32 × 2
Shape of $E_{t}$	81 × 81	-
Shape of $\varepsilon_{t}$	20 × 1	-

**Table 3 sensors-20-04574-t003:** Grid search for different combination of multiple channels.

Recent	Daily	Weekly	MAE	MAPE (%)	MdAE
5	3	3	14.29	20.32	7
5	3	1	14.46	20.60	7
5	4	2	15.01	21.05	7
5	2	2	14.68	20.96	7
6	3	2	14.74	20.90	7
4	3	2	14.80	20.96	7
**5**	**3**	**2**	**13.80**	**19.99**	**7**

**Table 4 sensors-20-04574-t004:** Baseline Component Description.

Method	Multi-Channel	Temporal	Spatial	External	Multi-Task
ARIMA		**√**			
STARIMA		**√**	**√**		
LSTM		**√**	**√**	**√**	
GeoMan		**√**	**√**	**√**	**√**
STNN		**√**	**√**		**√**
DMVST-Net		**√**	**√**	**√**	
ST-ResNet	**√**	**√**	**√**	**√**	
ADST-Net	**√**	**√**	**√**	**√**	**√**

**Table 5 sensors-20-04574-t005:** Prediction Performance Comparison on Two Public datasets.

Method	MetroHZ			TaxiBJ		
	MAE	MAPE (%)	MdAE	MAE	MAPE (%)	MdAE
ARIMA	92.09	88.31	50	28.09	93.25	13
STARIMA	61.21	58.24	30	26.04	61.45	14
LSTM	34.56	42.13	19	24.72	62.73	14
GeoMan	29.41	23.55	28	24.12	57.28	20
STNN	35.73	85.35	16	34.05	82.70	23
DMVST-Net	18.50	21.93	8	31.77	68.11	20
ST-ResNet	15.82	21.65	8	9.67	24.38	4
ADST-Net	**13.28**	**17.14**	**6**	**4.44**	**4.84**	**1**

**Table 6 sensors-20-04574-t006:** Prediction Performance at Multiple Time Steps (U = 5).

Method	MAE	MAPE (%)	MdAE
ARIMA	96.00	88.63	56
STARIMA	82.81	78.66	38
LSTM	46.02	47.68	19
GeoMan	68.96	66.94	54
STNN	44.29	60.22	41
DMVST-Net	72.28	71.66	34
ST-ResNet	74.53	72.65	30
**ADST-Net**	**37.94**	**33.38**	**16**

**Table 7 sensors-20-04574-t007:** The evaluation results of Self-comparison experiments.

Removed Component	MAE	MAPE (%)	MdAE
Attention Mechanism	13.65	17.30	6
Multi-scale convolution	13.79	17.81	7
Auxiliary task	13.85	17.70	6
External information	13.45	17.99	6
Multiple channels	14.00	17.90	6
ADST-Net	**13.28**	**17.14**	**6**

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
