# Peer review of "ADST: Forecasting Metro Flow Using Attention-Based Deep Spatial-Temporal Networks with Multi-Task Learning"

_sensors, 2020, doi:10.3390/s20164574_

Round 1

Reviewer 1 Report

The paper is valuable and well-presented, thus it can be accepted.

Reviewer 2 Report

this is a nice piece of paper that proposes a sound scientific approach to a timely problem. Big data problems and analysis are at the core of many current real-cases and techniques able to handle such problems are welcome. But not everything is such welcome, we (and the society) need new methods lying at the edge of the fields of computation and probability. This paper lies more on the computational aspect but slightly touches upon uncertainty. All in all, I would suggest a minor revision, close to acceptance.

These are some coments:

- although the written english is quite good, there are yet soe typos and bad use of the english. Please check it again.

- A paper is far more interesting when it provides code and the results can be reproducible. Would it be possible to make the code free, by adding a link for downloading?

- The paper stands on the machine learning - neural network approaches. This is fine but the paper would gain in interest if there would be a section on handling uncertainty within the proposed techniques. Unexpected things happen, and noisy observations can be at the core of the training. I would like to see some development under these cases.

- The spatial part of the approach is not clearly stated, at least as much clear as the temporal part. Please be more clear in this respect.

- Forecasting: the authors use four future temporal instants for prediction. What happens even further distant, what are the limits of the approach? 

Reviewer 3 Report

This study developed a new Attention-based deep spatial-temporal network with multi-task learning architecture (ADST-Net) for traffic flow prediction. The authors compared their proposed method to a few commonly used methods for traffic prediction.

The idea of incorporating spatial-temporal correlation and contextual information to improve the performance of machine learning prediction is not entirely new. Still, the authors’ method seemed to work very well with their testing data and has the potential of becoming something useful in the field. I do have a few suggestions to improve the presentation of this work.

First, the language is not bad (i.e., it has few grammatical errors), but the sentence structures and word choices often seem odd. I would suggest having a native speaker proofread the paper and restraining from using editing services ran by non-native speakers.

The literature review should include more discussions on different types of transportation big data (e.g., floating car data, metro card data, etc.).

In my opinion, Section 3 should be part of the methodology instead of being its own section, but I will leave this up to the authors.

There are a few details to clarify in the methodology, for example:

Table 1, row 4, why does number 2 represent the # of properties of metro flow Xt?

P6 l234, what do you mean by “temporal pair-wise relationship”? Please be concise in your word choice.

P7 l238, please clarify how you determined Q based on cross-validation?

Section 4.4,  how did you decide which external information to include in your model? There are a lot of external attributes to choose from (e.g., social-economic factors, weather, etc.)

P9 l 340, what do you mean by a “larger distribution”?

P15 l538, how did you decide the convolution kernel sizes?

I appreciate that the authors compared their methods with a few existing ones, but I find the comparison with ARIMA not very useful. There is a spatial version of ARIMA called ST-ARIMA; the authors should have compared their method with this one because ST-ARIMA also considers both spatial and temporal autocorrelation.

After the results section, I’d like to see a more thorough discussion of the limitations and potentials of their proposed method. For example, does it work with more sparsely distributed data such as LBSM data?

The Conclusion section is very short and reads more like a quick summary. It does not highlight the critical findings from the analysis.

Reviewer 4 Report

Please refer to the attached PDF document for my comments

Round 2

Reviewer 3 Report

The authors have addressed most of my comments satisfactorily, and I appreciate their effort to improve the manuscript.

Here is one additional minor comment:

P9 l334, the authors argued that “there are many external factors that may influence the flow prediction. However, among them, the rush hours and weather conditions play the main role in flow prediction.” ---- Please add citations to support your claim.

Other than that, I think the manuscript is ready for publication.
